# Proteomics-Driven Biomarkers in Pancreatic Cancer

**DOI:** 10.3390/proteomes11030024

**Published:** 2023-08-07

**Authors:** Luís Ramalhete, Emanuel Vigia, Rúben Araújo, Hugo Pinto Marques

**Affiliations:** 1Blood and Transplantation Center of Lisbon—Instituto Português do Sangue e da Transplantação, Alameda das Linhas de Torres, n° 117, 1769-001 Lisbon, Portugal; 2Nova Medical School, Faculdade de Ciências Médicas, Universidade NOVA de Lisboa, 1169-056 Lisbon, Portugal; 3iNOVA4Health—Advancing Precision Medicine, RG11: Reno-Vascular Diseases Group, NOVA Medical School, Faculdade de Ciências Médicas, Universidade NOVA de Lisboa, 1169-056 Lisbon, Portugal; 4Centro Hospitalar de Lisboa Central, Department of Hepatobiliopancreatic and Transplantation, 1050-099 Lisbon, Portugal; 5CHRC—Comprehensive Health Research Centre, NOVA Medical School, 1150-199 Lisbon, Portugal

**Keywords:** pancreatic cancer, proteomics, biomarker, chemotherapy effectiveness

## Abstract

Pancreatic cancer is a devastating disease that has a grim prognosis, highlighting the need for improved screening, diagnosis, and treatment strategies. Currently, the sole biomarker for pancreatic ductal adenocarcinoma (PDAC) authorized by the U.S. Food and Drug Administration is CA 19-9, which proves to be the most beneficial in tracking treatment response rather than in early detection. In recent years, proteomics has emerged as a powerful tool for advancing our understanding of pancreatic cancer biology and identifying potential biomarkers and therapeutic targets. This review aims to offer a comprehensive survey of proteomics’ current status in pancreatic cancer research, specifically accentuating its applications and its potential to drastically enhance screening, diagnosis, and treatment response. With respect to screening and diagnostic precision, proteomics carries the capacity to augment the sensitivity and specificity of extant screening and diagnostic methodologies. Nonetheless, more research is imperative for validating potential biomarkers and establishing standard procedures for sample preparation and data analysis. Furthermore, proteomics presents opportunities for unveiling new biomarkers and therapeutic targets, as well as fostering the development of personalized treatment strategies based on protein expression patterns associated with treatment response. In conclusion, proteomics holds great promise for advancing our understanding of pancreatic cancer biology and improving patient outcomes. It is essential to maintain momentum in investment and innovation in this arena to unearth more groundbreaking discoveries and transmute them into practical diagnostic and therapeutic strategies in the clinical context.

## 1. Introduction

Pancreatic ductal adenocarcinoma (PDA) is an invasive tumor with similar incidence and mortality rates. The incidence of PDA has increased worldwide in recent decades and is expected to continually rise [1,2,3,4,5], already being listed as the seventh leading cause of mortality by cancer worldwide [6].

By 2040, a 61.7% increase is expected in the total number of global cases [7]. The main difference between PDA and other cancers lies in the genomic heterogeneity of the tumors, which points to patient-specific tumoral genomic signatures [8]. This may explain the obstacles that prevent the identification of patient phenotypes predictive of better or worse prognosis. For instance, based on transcriptome analysis, the International Cancer Genome Consortium currently divides PDA into three molecular subtypes: progenitor, squamous, and aberrantly differentiated endocrine/exocrine types [9]. However, the decision from a multidisciplinary team, e.g., whether a specific patient with resectable disease would benefit from surgery, continues to be based on clinical information, laboratory tests, and imaging. In recent years, proteomics has emerged as a powerful tool for advancing our understanding of pancreatic cancer biology and identifying potential biomarkers and therapeutic targets [10].

Radical resection is the only potentially curative treatment for PDA [11]. Nonetheless, even after curative resection, up to 80% of patients experience disease relapse, resulting in a 5-year survival rate of only 20–30% [12]. Pancreaticoduodenectomy is associated with high morbidity (up to 60%) and an acceptable mortality rate below 5%, which strongly impacts both the patient’s quality of life and health costs [13]. Several studies and clinical trials have attempted to identify new biomarkers to improve therapies and formulate new healthcare policies. Major hurdles have been revealed in the diversity of disease phenotypes and in the costs of implementing new methodologies [14].

This narrative review on the application of proteomics-driven biomarkers in pancreatic cancer is aimed to provide an overview of the current state of the art and discuss the potential of proteomics in the early detection, diagnosis, and treatment of pancreatic cancer. For less familiar readers, this review also touches on other topics, such as why we should use proteomics to the detriment of other omics, as well as a round up of the major proteomics-driven methodologies that can or are applied to this subject of interest.

## 2. Screening and Diagnostic Accuracy: Where Do We Stand?

Pancreatic cancer diagnosis currently relies on imaging methods like CT, MRI, US, PET, and EUS, with detection sensitivities of 74%, 79%, 67%, 90%, and 94%, respectively [15]. However, these techniques lack significant prognostic value and do not guide therapeutic solutions [16,17]. Imaging inconsistencies occur due to variable interpretation and subtle early-stage cancer characteristics [17,18]. Emerging technologies like the advent of Artificial Intelligence (AI), machine learning (a branch of AI), and radiomics have been explored to address these issues [19]. These could potentially complement fluid-based biomarkers in early PDAC diagnosis, as seen in promising but limited studies [20]. Extensive research is underway to develop effective biomarkers for early detection, prognosis evaluation, and treatment response monitoring to improve the survival rate of PDA patients [21].

When it comes to laboratory discoveries linked to pancreatic cancer, the landscape is regrettably barren. Nowadays, there is still a necessity for the identification of more reliable and specific biomarkers for pancreatic cancer diagnosis. Biomarkers, such as carbohydrate antigen 19-9, also known as sialyl-Lewis (CA19-9), a carbohydrate antigen overexpressed in pancreatic cancer cells, have been widely used. However, this biomarker lacks both sensitivity and specificity, 80.8% and 89.1%, respectively [22,23], and with the predictability of resectability with an AUC of 0.72 for a cutoff value of 92.77 U/mL [24]. One other biomarker, also found to be wanting, is Mucin-16, also known as ovarian cancer-related tumor marker CA125 (MUC-16), a transmembrane glycoprotein that is overexpressed in pancreatic cancer. This last one though has a better track record in forecasting the resectability of pancreatic cancer, with an AUC of 0.81 versus 0.66 compared to CA19-9 [25]. Another biomarker is the carcinoembryonic antigen (CEA), a glycoprotein usually produced by normal cells during embryonic development, becoming undetectable or in very low concentrations in normal adult conditions. However, in situations where inflammatory processes manifest themselves and/or in the presence of tumors of the gastrointestinal tract, it tends to increase [26]. Despite most research on CEA and its potential for predicting outcomes in pancreatic cancer involving only a small number of patients, a study by Lee et al. found that CEA could be a promising biomarker, particularly for those patients for whom surgery is not an option [27]. Similarly, Tas et al. observed that patients with normal levels of LDH, CA19-9, and CEA had a better outcome than other biomarkers combination, and normal values of LDH and CEA were also associated with better survival [28].

## 3. What Can We Do Differently?

In medical research, all branches of molecular biology, the so-called “omics” (proteomics, genomics, and transcriptomics, among others), have played their role in the understanding of pancreatic cancer, as they study different aspects of biology at the molecular level. These different approaches are particularly important in the case of oncological disease, as it is characterized by abnormal changes in cellular behavior (e.g., protein expression and protein activity) [29,30,31]. Each approach within the ‘omics’ fields offers unique insights into the biology of organisms, not making one technique necessarily superior to others. However, proteomics proves particularly useful in certain contexts due to its ability to measure protein expression, activity, protein–protein interactions, signaling pathways, and post-translational modifications directly [32,33,34].

Proteomics is the study of proteins and their functions, contributing to the understanding of the underlying mechanisms of various biological phenomena. Recent advances in technology have led to the rapid growth and adoption of proteomics, already having provided remarkable contributions in the fields of biology, biotechnology, and medical research. In the medical field, these methodologies have benefited from a wide range of applications. They have been used for disease diagnosis and biomarker identification, which involves studying the levels or patterns of proteins in different biofluids. This allows for a deeper understanding of various diseases, the impact of individual proteins, and how their expression, structure, and function contribute to disease onset and progression or treatment. Furthermore, these techniques aid in the identification and validation of new therapeutic targets. They can also help ascertain drug efficacy and toxicity, providing fresh insights into the interactions between proteins and drugs [35,36,37]. Proteomics can also play a vital role in precision and personalized medicine. The knowledge of an individual’s proteome will allow one to tailor treatments to the specific needs and phenotype of an individual, increasing the chances of success for a given therapeutic course. This can be especially important in those cases where traditional treatments have failed, allowing for a more targeted approach to treatment [38].

However, despite the many benefits that proteomics presents, there are also some challenges. One of the biggest challenges lies in the complexity of the proteome as they can be affected by several of the following factors:The complexity of biofluid protein contents: biofluids obtained from individuals have more than one protein, sometimes in the range of thousands, interacting with each other, which can simultaneously be expressed in different isoforms [39].Interactions: proteins interact with each other to form complex networks that regulate cellular processes. In the event of a disease, these interactions can be disrupted, leading to abnormal cellular behavior [40].Diversity: proteins are diverse in their structure and function, which is reflected in their involvement in different diseases [41].Post-translational modifications: proteins can undergo a variety of post-translational modifications, such as phosphorylation, acetylation, and ubiquitination, which can dramatically alter their activity and localization. These modifications can be critical in the development and progression of disease [42].Dynamics: proteins are dynamic molecules that undergo continual changes in their expression and activity. When a disease is in effect, these changes can be rapid and profound, making it difficult to understand the underlying molecular mechanisms [43]. Current proteomic techniques are also often limited in their ability to accurately measure the levels of individual proteins [44].

As described before, there are many omics in existence, with some more known to the overall scientific community, whereas the rest is often kept out of the spotlight. Table 1 highlights a more encompassing view of omics and what they can or cannot do and directly compares them with proteomics.

## 4. Which Proteomic Technique and Sample Type Should Be Used?

The choice of proteomic methods is highly dependent on several critical factors that shape the research objectives and experimental design, e.g., specific study topics, sample characteristics, available resources, the experience of the research team, and the type of proteomic analysis desired. For instance, if the goal is to identify novel biomarkers for early pancreatic cancer detection, researchers might focus on highly sensitive methods such as Isobaric tags for relative and absolute quantitation (iTRAQ) [65] or Tandem Mass Tag (TMT) with Liquid Chromatography–Tandem Mass Spectrometry (LC-MS/MS) to compare protein expression levels in cancer and healthy tissues [66]. On the other hand, if the objective is to understand the underlying molecular mechanisms of pancreatic cancer progression, shotgun proteomics and RPPA (Reverse-Phase Protein Array) might be more suitable for studying the alterations in signaling pathways and protein networks [67].

In some cases, researchers might have limited access to patient samples, and thus, they need highly sensitive and low-input proteomic methods to maximize data output [68,69,70]. On the other hand, if ample tissue samples are available, besides the beforementioned iTRAQ, Shotgun Proteomics, or 2D Gel Electrophoresis and mass spectrometry, it is possible to explore multiple proteomic techniques, leading to a more comprehensive analysis [71,72].

Combining or integrating several proteomic approaches, although challenging, can enable a more thorough or holistic understanding of the pancreatic cancer proteome and signaling cascades, assisting in the development of novel treatment targets and biomarkers. Nevertheless, researchers that take this path must always consider the fact that different proteomic methods generate diverse types of data that require specialized analysis tools and pipelines.

As stated, proteomic techniques used in pancreatic cancer research can include 2D gel electrophoresis and mass spectrometry-based approaches such as LC-MS/MS, and iTRAQ, among others. Despite that, one of the most used techniques for pancreatic cancer research is LC-MS/MS, as it allows for the analysis of large numbers of proteins and their modifications in a single experiment, providing comprehensive and quantitative information about the proteome [73]. This method is particularly useful for identifying differentially expressed proteins in pancreatic cancer, as well as identifying protein–protein interactions, post-translational modifications, and protein expression levels.

For those curious or looking to brush up on existing proteomics methodologies and techniques, Table 2 provides a concise summary of the most used proteomics techniques or variations thereof, many of which are frequently employed in pancreatic cancer studies, either individually or in combination. This table, organized by the date of introduction into laboratories, outlines each method’s underlying principle, along with its key advantages and potential issues associated with its application. Understanding the strengths and limitations of these proteomic approaches is crucial for designing effective experimental strategies and advancing our knowledge toward improved diagnostics and targeted therapies for pancreatic cancer. Despite the beforementioned, many others can be used, including Enzyme-Linked Immunosorbent Assay (ELISA) [74,75,76] or Western blotting [77]. In the case of ELISA and due to its characteristics, namely, high sensitivity and specificity, quantitative nature, ease of use, and suitability for high-throughput screening, ELISA is an essential technique for protein analysis and biomarker discovery. In its many forms, either direct ELISA [78], indirect ELISA, sandwich ELISA [79], or ELISpot [80], ELISA can and is frequently used as an independent validation method for mass spectrometry-based proteomic data. In the case of Western blotting, also known as immunoblotting, it offers several of the following advantages in molecular biology and proteomics research: specificity (detection of a single protein using target-specific antibodies), sensitivity (suitable for studying proteins with limited expression), ease of use, usefulness in protein size determination, capability of quantification (providing semi-quantitative or quantitative data on relative protein abundance between samples), and broad applicability in terms of sample type (cells, tissues, and bodily fluids) [81,82,83]. Nevertheless, and despite its advantages, whenever using this method, it is crucial to be mindful of potential limitations, which may include nonspecific binding and antibody cross-reactivity or even the lack of appropriate controls to ensure the accuracy and reliability of the results.

In terms of biological samples used in these studies, there is no right answer, as the sample type will be highly dependent on several factors, namely, the scope of the study and sample availability. It is important to note that tumor tissue samples are crucial in pancreatic cancer research as they allow researchers to study the molecular changes that occur in the cancer cells, including genetic mutations, protein expression levels, and post-translational modifications. These molecular changes are critical in understanding the biology of pancreatic cancer and can help identify new therapeutic targets. In addition, tumor tissue samples are also important for the development of diagnostic biomarkers, which can be used to detect pancreatic cancer early and monitor its progression [84,85]. Indeed, tumor tissue contains the highest concentrations of proteins specifically associated with pancreatic cancer. Tumor tissue can be used with a variety of proteomics techniques, including 2-DE, MS, LC, MALDI, ICAT, SILAC, IAP, TMT, and iTRAQ. There are also other types of samples that may be used for pancreatic cancer protein studies. Peripheral blood mononuclear cells (PBMCs) are an important biological sample in pancreatic cancer research.

PBMCs are a type of immune cell that plays a crucial role in the body’s immune response to cancer. The study of PBMCs can provide valuable information about the systemic changes that occur in response to the disease, including changes in cytokine expression levels, the activation of immune cells, and changes in the transcriptome. This information can help researchers understand the role of the immune system in pancreatic cancer and identify new therapeutic targets [86,87].

Serum or plasma samples are also commonly used in pancreatic cancer research as they provide a snapshot of the overall protein expression changes in the blood. Proteins in the blood can act as biomarkers for pancreatic cancer, and their levels can be measured to monitor the progression of the disease or the effectiveness of treatments. In addition, changes in the levels of specific proteins in the blood can also indicate the presence of other related diseases, such as liver or bile duct cancer [88,89].

Another interesting biofluid is pancreatic juice, which, in recent years, has become an important tool in the study of pancreatic cancer. This biofluid has been particularly interesting as it presents itself as a non-invasive means of examining the pancreas. When analyzed via proteomics, this fluid can provide valuable information about the early stages of pancreatic cancer, proving to be a useful tool for monitoring pancreatic cancer progression and treatment efficacy [90,91], whereas other diagnostic methods are not sensitive enough to detect the disease.

**Table 2 proteomes-11-00024-t002:** Main proteomics techniques or variations, presented by date of the techniques’ introduction (from the top—earliest—to the bottom of the table—most recent—and including the principle of each technique alongside their major advantages/limitations).

Technique (Acronym)	Principle	Advantages	Limitations	Ref.
Enzyme-Linked Immunosorbent Assay (ELISA)	Proteins are immobilized on a plate and then probed with an antibody specific to the protein of interest. The amount of antibody bound to the protein is then measured.	Sensitive and quantitative, can be used to identify and quantify proteins.	The sensitivity of ELISA can be affected by the presence of contaminants in the sample and the narrow dynamic range.	[92,93]
Two-dimensional gel electrophoresis (2DE)	Proteins are separated by their molecular weight in the first dimension and by their isoelectric point in the second dimension.	High resolution and sensitivity, can be used to identify protein post-translational modifications (PTMs).	Can be difficult to identify proteins that are very similar in size or charge and low throughput.	[94,95,96]
Western Blot	Proteins are separated by 2DE or SDS-PAGE and then transferred to a membrane. The membrane is then probed with an antibody specific to the protein of interest.	Sensitive and specific, can be used to identify and quantify proteins.	The sensitivity of Western blot can be affected by the presence of contaminants in the sample and antibody specificity.	[97,98]
Capillary electrophoresis (CE)	Proteins are separated by their size and charge in a capillary.	Sensitive and high-throughput, can be used to study protein–protein interactions.	The resolution of CE can be lower than that of 2DE.	[99,100]
Mass spectrometry (MS)	Proteins are ionized and fragmented, and the resulting fragments are analyzed by their mass-to-charge ratio.	High sensitivity and accuracy, can be used to identify and quantify proteins and to study PTMs.	Can be difficult to identify proteins that are very similar in mass.	[101,102,103]
Liquid chromatography–mass spectrometry (LC-MS)	Proteins are separated via liquid chromatography and then analyzed via MS. This allows for the identification and quantification of a wide range of proteins.	Sensitive and quantitative, can be used to study protein–protein interactions and PTMs.	The complexity of LC–MS can make it difficult to interpret the results.	[104,105,106]
Isotope-coded affinity tag (ICAT)	Proteins are labeled with different isotopes before being separated via MS. This allows for relative quantitation of proteins.	Sensitive and quantitative, can be used to study protein turnover.	The isotopes used in ICAT can be expensive.	[107,108]
2D Differential gel electrophoresis (DIGE)	Modification of 2DE. Proteins are labeled with different fluorescent dyes before being separated via 2DE. This allows for the visualization of changes in protein expression between two samples.	Sensitive and quantitative, can be used to study protein expression and overcomes limitations in traditional 2DE.	The dyes used in DIGE can be expensive.	[109,110]
Peptide mass fingerprinting (PMF)	Proteins are digested into peptides, and the masses of the peptides are determined using MS (e.g., MALDI-TOF or ESI-TOF). This allows for the identification of proteins.	Sensitive and relatively inexpensive, can be used to identify proteins in complex mixtures.	The accuracy of PMF can be affected by the presence of contaminants in the sample.	[111,112,113]
Protein microarrays	Proteins are immobilized on a chip before being probed with antibodies or other molecules. This allows for the identification and quantification of proteins that interact with the probes.	High-throughput and multiplexing can be used to study protein–protein interactions.	The complexity of protein microarrays can make it difficult to interpret the results.	[114,115]
Surface-enhanced laser desorption/ionization time-of-flight mass spectrometry (SELDI-TOF MS)	Variation of MALDI-TOF. Proteins are immobilized on a surface before being analyzed via MS. This allows for the identification of proteins that interact with the surface.	Sensitive and specific, can be used to study protein–protein interactions.	The specificity of SELDI-TOF MS can be affected by the presence of contaminants in the sample. Problematic in detecting larger MW proteins and PTM.	[116,117,118]
Stable isotope labeling with amino acids in cell culture (SILAC)	Proteins are labeled with different isotopes during cell culture. This allows for relative quantitation of proteins.	Sensitive and quantitative, can be used to study protein turnover and protein–protein interactions.	The isotopes used in SILAC can be expensive.	[119,120]
Isobaric tags for relative and absolute quantitation (iTRAQ)	Proteins are labeled with different isobaric tags before being separated via MS. This allows for both relative and absolute quantitation of proteins.	Sensitive, quantitative, and versatile, can be used to study protein turnover and protein–protein interactions.	The isobaric tags used in iTRAQ can be expensive.	[121,122]
Label-free quantitative proteomics	Proteins are separated via MS without the use of labels. This allows for absolute quantitation of proteins.	Sensitive and quantitative, does not require the use of specialized equipment. Achieves high-proteome coverage and simpler workflows. Variability of chemical labeling/tagging is eliminated.	The accuracy of label-free methods can be affected by the presence of contaminants in the sample.	[123,124,125]
Multidimensional protein identification technology (MudPIT)	Proteins are separated by two or more dimensions of liquid chromatography before being analyzed via MS. This allows for the identification of a wider range of proteins.	Sensitive and comprehensive, can be used to study protein–protein interactions and PTMs.	The complexity of MudPIT can make it difficult to interpret the results.	[126,127]

ELISA, Enzyme-Linked Immunosorbent Assay; 2DE, Two-dimensional gel electrophoresis; PTMs, Post-translational modifications; MALDI-TOF, Matrix-assisted laser desorption/ionization; ESI-TOF, Electrospray Ionization Time-of-Flight; SDS-PAGE, sodium dodecyl sulfate–polyacrylamide gel electrophoresis; CE, Capillary electrophoresis; LC–MS, Liquid chromatography–mass spectrometry; MS, Mass spectrometry; ICAT, Isotope-coded affinity tag; DIGE, Differential gel electrophoresis; PMF, Peptide mass fingerprinting; SELDI-TOF MS, Surface-enhanced laser desorption/ionization time-of-flight mass spectrometry; SILAC, Stable isotope labeling with amino acids in cell culture; iTRAQ, Isobaric tags for relative and absolute quantitation; MudPIT, Multidimensional protein identification technology.

A significant advantage of proteomics in biomarker identification is its ability to provide early disease indicators, often before any symptoms become evident. This can be especially important in the case of pancreatic cancer, which is often asymptomatic in its early stages and, therefore, difficult to detect. By identifying changes in protein expression in the blood or other biological samples, proteomic biomarkers can be used to detect the disease at an earlier stage and improve the chances of successful treatment. It is crucial to remember that pancreatic cancer is frequently inoperable, as it is typically diagnosed at an advanced stage. The tumor’s location and growth pattern often make it challenging to entirely remove the tumor without causing damage to the surrounding tissue [128]. This highlights the importance of early detection and the development of new diagnostic methods to improve the chances of successful treatment.

This review provided an overview of different proteomics approaches focused on the identification of new diagnostic, prognostic, and predictive biomarkers, and the utility of these approaches in the identification of proteome signatures associated with treatment response in pancreatic cancer.

## 5. Proteomics as a Biomarker Source for Pancreatic Cancer

Pancreatic cancer is a devastating disease that arises in the pancreas. It is one of the most lethal types of cancer, as it is often diagnosed at a late or advanced stage, resulting in outcomes with a poor prognosis [129]. As a result, there is a need for new diagnostic tools for early detection and prognosis of this cancer. Recent advances in proteomic analysis have led to the identification of numerous biomarkers for the diagnosis, early detection, prognosis, and classification of pancreatic cancer, providing valuable insights into the disease. To provide an answer to these challenges, a range of proteomic studies have been carried out to detect specific proteins and extracellular vesicles (EVs) that are differentially expressed in pancreatic cancer.

For instance, Jia et al. (2020) [130] employed iTRAQ-based analysis to identify differential serum proteins (RAD50, TGF-β1, and APAF1) that serve as diagnostic markers of pancreatic ductal adenocarcinoma. Meanwhile, Wu et al. (2021), also utilizing iTRAQ and mass spectrometry, identified three proteins (PROZ, TNFRSF6B, and TNFRSF6B) that, when combined, could provide an AUC of 0.932 for early-stage pancreatic cancer detection [131].

As previously stated, several studies have focused on the proteomic analysis of extracellular vesicles produced by cancerous versus healthy pancreatic organoids. EVs are small vesicles secreted by cells that contain proteins, nucleic acids, and other molecules that can be used as biomarkers. These particles have been identified as being implicated in cellular transformation in several cancer types, with pancreatic cancer not being an exception. In fact, several researchers have pointed out that EVs contribute to the initialization of malignant cell transformation [132,133].

One study found that the proteomic analysis of EVs could distinguish cancerous from healthy pancreatic organoids with high sensitivity and specificity, with tumor-promoting candidates, LAMA5, SDCBP, and TENA consistently upregulated in PDAC-derived EVs [134].

In a study published in 2021, the researchers used iTRAQ-based analysis of plasma-derived exosoma-identified ALG-2 interacting protein X (ALIX) as a novel biomarker for the diagnosis and classification of pancreatic cancer. The researchers reported an AUC of 0.91, with a 90.6% sensitivity and an 83.9% specificity for this marker when combined with CA 19-9 [135]. This protein had already been described as a regulator of both epidermal growth factor receptor (EGFR) activity and programmed death-ligand 1 (PD-L1 or CD 274) surface marker, indicating its involvement or regulatory capability in tumor-mediated immunosuppression [136,137], with its use as a biomarker being pointed out in other tumors, such as oral squamous cell carcinoma [138] or colorectal carcinoma, among others [137].

It has also been discovered that circulating cancer-associated EVs, derived from the serum of PDAC patients, could be used as early detection and recurrence biomarkers for pancreatic cancer [139]. In this study, two novel biomarkers were identified, G protein-coupled receptor class C group 5 member C (GPRC5C) and epidermal growth factor receptor pathway substrate 8 (EPS8), that enabled the discrimination between healthy controls and early-stage PDAC, with AUC values of 0.946 when combined with each other. One study conducted by Chen et al. (2023) used iTRAQ-based analysis to identify differential plasma proteins, which could serve as diagnostic markers for pancreatic ductal adenocarcinoma. This study found that three proteins, when combined with CA19-9, AAT, RAB2B, and IGFBP2, resulted in an AUC of 0.90, indicating a high diagnostic accuracy [140].

Although several articles have identified pancreatic cancer biomarkers, the presented results do not always lead to clear identification of the biomarkers, in part due to the molecular complexity of the disease. In some situations, instead of a specific biomarker, several authors have investigated protein patterns. In a recent study, Son et al., using reaction monitoring–mass spectrometry, identified 24 proteins that could classify patient outcomes in four risk subgroups, thus providing clinicians with new tools to identify high-risk patients who could benefit from more aggressive treatment [141]. On a similar note, Kafita et al., performing proteogenomic analyses of pancreatic cancer subtypes, identified subtype-specific protein expression patterns and genetic alterations, including alterations in pathways related to cell cycle regulation, DNA damage repair, and immune response. The researchers also identified potential therapeutic targets, including several protein kinases and immune checkpoint molecules. These researchers have also identified potential therapeutic targets that could have important implications for the development of personalized treatments for pancreatic cancer patients. Of the two types of pancreatic tumors that the authors identified through machine learning, namely subtype-1 and subtype-2, a comparison was made regarding the expression levels of various proteins between the two disease subtypes. The discovery revealed that subtype-1 tumors displayed significantly elevated expression levels of multiple proteins, such as mTOR, E-Cadherin, and Raf-pS338, in contrast to subtype-2 tumors, which manifested significantly increased expression levels of proteins, including Stathmin, Mre11, and MAP2K1 [142]. Silwal-Pandit et al., using LC-MS, highlighted the importance of the extracellular matrix in pancreatic cancer progression, suggesting that extracellular matrix proteins could serve as potential prognostic biomarkers for pancreatic cancer patients. Further analysis found that an elevated expression of several proteins network involved in epithelial–mesenchymal transition and glycolytic activities, low oxidative phosphorylation, E2F, and DNA repair pathway activities [84].

In addition to the biomarkers for pancreatic cancer previously presented, there is a wealth of research focused on identifying other potential biomarkers for this disease. In Table 3, a list of articles related to novel biomarkers for pancreatic cancer can be consulted. These articles cover a wide range of new, potential, and reliable biomarkers for pancreatic cancer, identified in different biological samples, crucial for improving patient outcomes and advancing our understanding of this devastating disease.

In summary, the use of proteomics in the identification of biomarkers for pancreatic cancer has the potential to improve discrimination, early detection, and prognosis of this afflicting disease. Regardless, further studies are needed to validate these biomarkers and translate them into clinical use.

## 6. Proteomics Signatures Associated with Treatment Response

As previously noted, pancreatic cancer is an extremely aggressive form of cancer with a grim prognosis. According to the American Cancer Society, the five-year survival rate for pancreatic cancer is only about 11% [159,160]. The limited treatment options for pancreatic cancer often include surgery, radiation therapy, and chemotherapy, but the outcomes of these treatments are generally unsatisfactory due to the aggressive nature of the disease [161]. As a result, there is an urgent need to identify new therapeutic targets and biomarkers for pancreatic cancer treatment.

Omics, which encompasses proteomics, has emerged as a powerful tool, for analyzing the global protein expression patterns of cancer cells. Proteomics can identify proteins that are differentially expressed between cancer and normal cells, and between different stages of pancreatic cancer, before, during, and after treatment, which can lead to the identification of new therapeutic targets and enable personalized cancer treatment. By applying these methodologies, the patient prognosis can be provided and help guide treatment decisions and predict drug-associated adverse events [162,163,164].

In the precision medicine or holistic medicine approaches, proteomic signatures or proteomic profiling have been researched in the context of pancreatic cancer treatment response and the patient’s overall outcome. For example, a study by Peng et al. identified a plasma proteomic signature associated with the response to chemotherapy in pancreatic cancer patients (vitamin-K dependent protein Z, sex hormone-binding globulin, von Willebrand factor, and CA 19-9). The authors suggested that this proteomic signature could be used to distinguish good responders from limited responders for stage III and stage IV patients with an AUC of 0.83 and 0.87, respectively [165]. One other study, conducted in tumor and adjacent pancreas tissue samples by Sahni et al., identified a group of 19 proteins (e.g., GRP78, CADM1, PGES2, and RUXF with AUC ≥ 0.92) that were significantly upregulated in poor-responders, enabling the prediction of chemo-resistant tumor phenotype [166]. Also, Le Large et al. [167], in a study conducted on gemcitabine-sensitive and gemcitabine-resistant cell lines, identified two proteins microtubule-associated protein 2 (MAP2) and anti-ankyrin-3 (ANK3), highly upregulated and phosphorylated in cell resistant cells. Kim et al., also working with cell lines, identified a panel of 107 proteins in which expression levels changed between oxaliplatin-resistant and sensitive cells. In this study, a stable isotope labeling by amino acids in cell culture (SILAC)-based quantitative proteomics analysis strategy, myristoylated alanine-rich C-kinase substrate (MARCKS) and WLS (Wnt/β-catenin signaling), was demonstrated to be involved in oxaliplatin resistance in pancreatic cancer cells [168]. Similarly, Chiu et al. also pointed out in a review paper, the use of MARCKS in the metastasis and treatment resistance of solid tumors [169]. In the case of the WLS, several studies have pointed in the direction of developing small-molecular compounds targeting the WLS pathway in disease treatment, as reviewed by Liu et al. [170]. In addition, Lin et al., in a study on the evaluation of gemcitabine resistance metabolomic profile, observed that many differentially expressed proteins quantified in mutant gemcitabine-resistant cells, revealing that these cells modulate several pathways to adapt to gemcitabine-induced stress. These authors also postulate that the therapeutic effectiveness could be increased by targeting the gemcitabine metabolic pathways with the introduction of treatment combinations, which would increase gemcitabine efficacy [171]. Gemcitabine, being one of the main chemotherapy drugs used to treat pancreatic cancer, has led other researchers to evaluate the resistance and sensitivity to this drug. For instance, Kim et al. [172], identified 13 epithelial to mesenchymal transition-related proteins which were closely associated with drug resistance and differentially expressed. In a more recent study, Amrutkar et al. [173] identified multifunctional cell types found in endocrine and exocrine pancreatic tissue, known as pancreatic stellate cells, that are quiescent and regulate extracellular matrix production and, from these cells, identified diverse protein expression profiles that could be associated with gemcitabine-resistance.

However, despite the very interesting results obtained by Le Large et al. [167], Kim et al. [168], and others, with cell lines, it is important to keep in mind that the complexity of an in vivo system is very different from an in vitro system. In fact, as demonstrated by Coleman et al. [174], some proteins expression can be lost in cell lines, e.g., 63 proteins were exclusively expressed in patient tissue samples, and 324 proteins were identified as specific to the cell line, which, in this case, most probably are proteins associated with cell survival in culture.

In summary, these studies underscore the promising potential of proteomic signatures in forecasting treatment responses and patient outcomes in pancreatic cancer. Proteomic analyses have yielded invaluable insights into the molecular mechanisms driving the onset, progression, and treatment responses of this lethal disease, thanks to an understanding of these mechanisms. This has led to the creation of new tools for prognosis and prediction based on proteomic signatures, offering hope to enhance clinical care for patients with pancreatic cancer.

## 7. Final Remarks

In conclusion, proteomics is an accelerating domain with the power to revolutionize our comprehension of pancreatic cancer biology and lay the groundwork for creating more effective diagnostic and treatment strategies. Through state-of-the-art techniques like mass spectrometry and bioinformatics, scientists can detect and quantify protein expression patterns associated with pancreatic cancer progression, treatment response, and prognosis.

Even though there is considerable work yet to be carried out and much more to discover, the application of proteomics in pancreatic cancer research has already provided encouraging results. Regarding screening and diagnostic precision, proteomics holds the potential to refine the accuracy of existing methods. By discovering novel biomarkers and therapeutic targets, proteomics could greatly enhance patient outcomes and ultimately pave the way for personalized medicine in pancreatic cancer treatment.

However, it is crucial to remember that numerous challenges must be confronted to fully exploit the power of proteomics in pancreatic cancer research, for instance, boosting the sensitivity and specificity of proteomic tests, as well as crafting standardized protocols for sample preparation and data analysis.

To tap into the full potential of proteomics in pancreatic cancer research, continuous investment in new technologies, methodologies, and innovative drug development is essential. Allocating resources to extensive patient multicenter studies will be vital in advancing our understanding of pancreatic cancer and enhancing patient outcomes. These large multicenter studies can play a key role in addressing some of these challenges as follows:Facilitating extensive data collection: multicenter studies, given their wider reach, can accumulate data from a vast number of patients. This offers invaluable insights into the intricacies of pancreatic cancer and how it reacts to various treatments.Broadening patient diversity: by including multiple centers, these studies can capture data from patients across varied geographies, demographics, and healthcare systems. This diversity ensures that the research conclusions have broader relevance.Bolstering statistical reliability: the larger the study size in multicenter research, the greater the statistical weight and confidence in the findings.Fostering collaboration and shared knowledge: bringing together various centers for these studies fosters a culture of cooperation and shared learning among researchers, potentially quickening the pace of innovations.Promoting protocol standardization: with multiple centers involved, there is an inherent push towards harmonizing protocols for sample collection, data interpretation, and other pivotal research processes. This is essential for ensuring consistency, reliability, and reproducibility in findings.

To summarize, while numerous challenges persist, proteomics possesses the immense potential to redefine our grasp on pancreatic cancer biology and improve patient outcomes. Sustained investment and continuous innovation in this domain are paramount for unveiling even more pivotal insights in the coming times.

## Figures and Tables

**Table 1 proteomes-11-00024-t001:** Comparative advantages of proteomics over other omics.

Omics	Advantage	Disadvantage	Possible Benefit of Using Proteomics over This Omic	Ref.
Genomics	Can identify genetic variations and mutations.	Does not reflect real-time cellular events.	Proteomics can reflect changes in protein levels and post-translational modifications, which provide a real-time snapshot of cellular events.	[45,46]
Transcriptomics	Can measure gene expression levels.	Does not reflect protein levels or activity.	Proteomics measures protein abundance and activity, which is the ultimate determinant of cell behavior and phenotype, as protein levels do not always correlate with mRNA levels.	[47,48]
Metabolomics	Provides insights into end products of cell metabolism.	Only captures final steps of cellular processes.	Proteomics offers a comprehensive look at the many steps involved in cellular function and disease processes, including the regulation and interaction of proteins.	[49,50,51]
Epigenomics	Studies heritable changes not coded in DNA sequence.	Limited in predicting functional outcomes.	Proteomics can indicate functional outcomes due to post-translational modifications and protein–protein interactions, which often depend on epigenetic changes.	[52,53]
Interactomics	Studies interactions and associations between proteins.	Limited in scale and often lacks context.	Proteomics can provide context by identifying abundance of proteins and can also explore protein modifications, adding depth to interaction data.	[54,55]
Phosphoproteomics	Identifies and characterizes phosphorylated proteins.	Limited to one type of post-translational modification.	Proteomics can identify many different types of post-translational modifications, offering a broader view of protein activity.	[56,57]
Glycomics	Studies the entire complement of sugars in an organism.	Technically complex and hard to interpret.	Proteomics can identify glycosylated proteins and help link these modifications to functional changes, providing insights into the role of sugars in biology.	[58,59]
Lipidomics	Targets and studies unique roles of lipids in organisms.	Does not directly link to protein function.	Proteomics can identify proteins that interact with lipids or are modified by them, providing functional context to lipidomics data.	[60,61,62]
Microbiomics or Metagenomics	Studies genetic material in a microbiome.	Does not reflect the impact of the host’s proteins.	Proteomics can study how host proteins interact with and are affected by the microbiome, offering insights into host–microbe interactions.	[63,64]

**Table 3 proteomes-11-00024-t003:** Proteomics studies on pancreatic cancer are conducted on plasma, serum samples, biopsy tissue, or PBMC, pointing to the analytical technique used, the population phenotype, dimension, and the predictive model’s output. The studies were sequenced according to year of publication (from the oldest—top—to the earliest—bottom).

Biofluid Type Proteomic (Technique)	Population Dimension (It Is Indicated If an Independent Validation Set Was Used)	Prediction Models (Peptide Fragments/Proteins Used in the Model)	Ref.
Tissue (Tissue microarray)	140 PDAC	(Galectin 4) for early recurrence	[143]
Cell Line (LC-MS-MS + WB)	PC-1.0 and PC-1 cell line	(T-complex protein 1 subunit theta) for cancer invasion and metastasis	[144]
Oral Fluid (2DE + MS)	15 PCP and 16 HC (10 PCP and 10 HC)	AUC = 0.91 with sensitivity and specificity of 90.0% (Cytokeratin-14, Lactoperoxidase, Cytokeratin-16, Cytokeratin-17, and Peptidyl-prolyl cis–trans isomerase B)	[145]
Plasma + plasma-derived microparticle (LC-MS-MS)	12 PCP	q < 0.1 (Receptor-type tyrosine-protein phosphatase um; Receptor-type tyrosine-protein phosphatase beta; 26S proteasome non-ATPase regulatory subunit 11)	[146]
Tissue (LC-MS-MS)	10 PDAC + 10 normal pancreatic biopsies	log2 fold change 6.4; *p* = 5 × 10^−6^ (Yes-associated protein 1)	[147]
Cell line (2D + MALDI-TOF MS)	PANC1 and PANC1-I5 cell line	Galectin-1	[148]
Secreted Extracellular vesicles (LC-MS-MS)	apan-2, MIA PaCa-2,Panc-1, and HPDE cell line	Peptide FDR < 0.01, protein FDR < 0.01 (348 proteins uniquely identified in cancer cell lines)	[132]
Plasma (LC-MS-MS)	22 PCP 15 good and 7 poor responders to post-neoadjuvant chemotherapy	R^2^ = 0.7 (Apolipoprotein A-IV, Ceruloplasmin, Complement C3 and Complement factor B, Complement C1q subcomponent subunit B, Complement C2, Complement C4-B, Transthyretin and Zinc-alpha-2-glycoprotein s)	[149]
Plasma (iTRAQ)	10 metastasis-free PCP + 10 PCP with distant metastasis + 10 gallstones (51 PC + 40 with gallstones)	AUC = 0.956 (SERPINA1 protein + CA19-9)	[150]
Plasma + Tissue (Proteome array)	14 PDAC	(23 proteins two-fold change)	[151]
Plasma (array-based technology)	135 PDAC + 72 HC + 13 benign lesions/chronic pancreatitis (75 PDAC + 36 HC + 19 chronic pancreatitis)	AUC = 0.89 (tyrosine-protein kinase Lyn, ITGB5, CEACAM1, secreted protein acidic and cysteine rich, alpha-taxilin, cyclin-dependent kinase inhibitor 1, annexin 1 and CA19-9)	[152]
Tissue (LC-MS-MS)	173 samples from 52 PDAC	(Mucin 5, GATA6)	[153]
Tissue (WB)	6 PDAC + 6 normal pancreatic biopsies	(Ubiquitin thioesterase OTU1)	[154]
Serum (HuProt microarray + ELISA)	338 PDAC + 294 HC + 122 chronic pancreatitis + 100 non-PDAC malignancies	AUC = 0.93 (CLDN17, KCNN3, SLAMF7, SLC22A11 and OR51F2)	[155]
Tissue (MALDI-MSI)	13 PDAC (primary + metastatic)	accuracy = 90% (COL1A1, COL1A2, and COL3A1)	[156]
Serum extracellular vesicles (MS/MS + immunoblotting)	15 PDAC + 25 HC + 14 pancreatitis Patients (27 PDAC + 7 HC + 8 pancreatitis Patients)	AUC = 0.89 (G Protein-Coupled Receptor Class C Group 5 Member C; Epidermal growth factor receptor kinase substrate 8)	[139]
Tissue + PBMC (MS)	12 PDAC + 2 chronic pancreatitis patients	(cell death protein 1)	[87]
Plasma (array-based technology)	610 PCP + 623 non-PCP	C statistic 0.779 (Monocyte chemotactic protein 3; Angiopoietin-2; Interleukin-18; Interleukin-6; Lysosome-associated membrane glycoprotein 3; C-C motif chemokine 3; CD4 T cell surface glycoprotein; T cell surface glycoprotein CD8 alpha chain; Heme oxygenase 1; Hepatocyte growth factor; Interleukin-2; Interleukin-4; Granzyme A; Cytotoxic and regulatory T cell molecule; Adhesion G-protein coupled receptor G1)	[157]
Serum (high-throughput proteomics dataset)	83 individuals at risk of PDAC	AUC = 0.9 (PCSK9, FGF-BP1, PLA2G7, LYPD3, and MSLN)	[158]

PDAC, Pancreatic ductal adenocarcinoma; LC-MS-MS, Liquid Chromatography–Tandem Mass Spectrometry; WB, Western blot; MS, mass spectrometry; MALDI-TOF MS, Matrix-assisted laser desorption/ionization time-of-flight mass spectrometry; iTRAQ, Isobaric tags for relative and absolute quantitation; PCP, pancreatic cancer patients; HC, healthy volunteers.

## Data Availability

Not applicable.

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
