# Peer review of "Proteomics-Driven Biomarkers in Pancreatic Cancer"

_proteomes, 2023, doi:10.3390/proteomes11030024_

Round 1
Reviewer 1 Report
The current review paper titled “Proteomics in Pancreatic Cancer”, is informative and discusses about the importance of proteomic techniques in identification of biomarkers of pancreatic cancer, and potentially be used for diagnosis, prognosis of the condition. Though the paper seems to be interesting, I have few suggestions and corrections, and the paper would benefit from additional work.
1. Line 19 – rewrite – “In this review article it is provided” with “This review article provides”.
2. Line 21-22. The whole sentence seems to be repetitive. Please re-write.
3. Line 68, reference numbering is missing for Gonoi et al.,
4. Line 76 – reference numbering is missing for Preuss et al.,
5. Section 4: Which proteomic techniques to use? – Authors had provided list of different techniques used in proteomics as a table. However, it would be very informative to readers if ELISA or other traditional proteomic techniques were included. Or authors only wanted to discuss LCMS analysis? And also, a lot of this section discussed on about the sample types used for the proteomic investigation. I suggest including in this in the section title.
6. Line 281 – include some examples of major extracellular matrix proteins reported in the study.
7. Line 332 – reference numbering is missing for Le Large et al.,
8. Over-all sentence structure could be improved and avoid extremely long sentences.
Overall English language is good. However, there are few grammatical errors and need some corrections.
Author Response
Dear Reviewer
Thank you for giving us the opportunity to submit a revised draft of our manuscript. We appreciate the time and effort that you and the other reviewers have dedicated to providing your valuable feedback on our manuscript. We are grateful for your insightful comments on our paper. We have been able to incorporate changes to reflect most of the suggestions provided by the reviewers. We have highlighted the changes within the manuscript.
Here is a point-by-point response to the reviewers’ comments and concerns.
R1
Comments and Suggestions for Authors
The current review paper titled “Proteomics in Pancreatic Cancer”, is informative and discusses about the importance of proteomic techniques in identification of biomarkers of pancreatic cancer, and potentially be used for diagnosis, prognosis of the condition. Though the paper seems to be interesting, I have few suggestions and corrections, and the paper would benefit from additional work.
- Line 19 – rewrite – “In this review article it is provided” with “This review article provides”.
- Line 21-22. The whole sentence seems to be repetitive. Please re-write.
- Line 68, reference numbering is missing for Gonoi et al.,
- Line 76 – reference numbering is missing for Preuss et al.,
Reply from authors: thank you for your suggestions, all points were corrected, and English used was made more succinct. This manuscript was edited for proper English language, grammar, punctuation, spelling, and overall style by one or more of the highly qualified native.
- Section 4: Which proteomic techniques to use? – Authors had provided list of different techniques used in proteomics as a table. However, it would be very informative to readers if ELISA or other traditional proteomic techniques were included. Or authors only wanted to discuss LCMS analysis? And also, a lot of this section discussed on about the sample types used for the proteomic investigation. I suggest including in this in the section title.
Reply from authors: All points were taken into consideration and added both in the form of a new table and further more specific discussion on section 4 of the manuscript.
- Line 281 – include some examples of major extracellular matrix proteins reported in the study.
- Line 332 – reference numbering is missing for Le Large et al.,
- Over-all sentence structure could be improved and avoid extremely long sentences.
Reply from authors: thank you for your suggestions, all points were corrected, examples of ECM proteins from the study were included.
Comments on the Quality of English Language
Overall English language is good. However, there are few grammatical errors and need some corrections.
Reply from authors: thank you for the comments, overall, this manuscript was edited for proper English language, grammar, punctuation, spelling, and overall style by one or more of the highly qualified native.

Reviewer 2 Report
The current review summarized the field of proteomics in diagnosis and treatment of pancreatic cancer. Overall the manuscript is factual and well written. However, it lacks depth and structure. Therefore I would not publish it in its current state.
Paragraph 1-3 in Section 2 are irrelevant to the topic and should be removed or summarized in a few sentences.
The authors did a great job listing some of the considerations when using proteomics to study cancer (Line 140-156). I would suggest the authors also list a table that describes the different omics methods and why proteomics offer unique advantages.
The authors detailed various proteomics methods in Section 4, in which a healthy discussion of the various pancreatic samples was mentioned. I would suggest to have the latter in a different section, and also highlight which proteomic methods might be most useful for which type of tissue samples.
The authors reviewed the current state of biomarker research in pancreatic cancer, the majority of which facilitated by proteomics. However, there is a lack of summarization and curation of these studies. I would suggest the authors discuss more of their opinions on this topic.
Finally, it was unclear to the reviewer what the scope of this manuscript is. If it is a broad discussion on proteomics in pancreatic cancer, then the authors should include more aspects of the disease biology and/or clinical studies that utilize proteomics methods other than biomarker detection. I would suggest the authors either include that or rename the review and make the manuscript more focused on utilizing proteomics in pancreatic cancer biomarker research.
The manuscript is well written and easy to follow. Minor grammar mistakes did not influence the reviewer understand the authors' main messages.
Author Response
Dear Reviewer
Thank you for giving us the opportunity to submit a revised draft of our manuscript. We appreciate the time and effort that you and the other reviewers have dedicated to providing your valuable feedback on our manuscript. We are grateful for your insightful comments on our paper. We have been able to incorporate changes to reflect most of the suggestions provided by the reviewers. We have highlighted the changes within the manuscript.
Here is a point-by-point response to the reviewers’ comments and concerns.
R2
Comments and Suggestions for Authors
The current review summarized the field of proteomics in diagnosis and treatment of pancreatic cancer. Overall the manuscript is factual and well written. However, it lacks depth and structure. Therefore I would not publish it in its current state.
Paragraph 1-3 in Section 2 are irrelevant to the topic and should be removed or summarized in a few sentences.
The authors did a great job listing some of the considerations when using proteomics to study cancer (Line 140-156). I would suggest the authors also list a table that describes the different omics methods and why proteomics offer unique advantages.
Reply from authors: the authors thank the reviewer for the suggestions, paragraphs 1-3 from section 2 were drastically reduced and the information summarized. The authors have also included a new table in which they describe advantages and disadvantages of the major Omics, as well as the counterpoint to the valuable use of proteomics.
The authors detailed various proteomics methods in Section 4, in which a healthy discussion of the various pancreatic samples was mentioned. I would suggest to have the latter in a different section, and also highlight which proteomic methods might be most useful for which type of tissue samples.
The authors reviewed the current state of biomarker research in pancreatic cancer, the majority of which facilitated by proteomics. However, there is a lack of summarization and curation of these studies. I would suggest the authors discuss more of their opinions on this topic.
Finally, it was unclear to the reviewer what the scope of this manuscript is. If it is a broad discussion on proteomics in pancreatic cancer, then the authors should include more aspects of the disease biology and/or clinical studies that utilize proteomics methods other than biomarker detection. I would suggest the authors either include that or rename the review and make the manuscript more focused on utilizing proteomics in pancreatic cancer biomarker research.
Reply from authors: thank you for the comments. There has been an extensive revision of section 4 to accommodate for the reviewer’s inputs.
Comments on the Quality of English Language
The manuscript is well written and easy to follow. Minor grammar mistakes did not influence the reviewer understand the authors' main messages.
Reply from authors: thank you for the comments, overall, this manuscript was edited for proper English language, grammar, punctuation, spelling, and overall style by one or more of the highly qualified native.

Reviewer 3 Report
Overall, the review provides a comprehensive analysis and valuable insights into the future utilization of proteomics in the field of pancreatic cancer. However, there are concerns regarding the accuracy and organization of Table 1, which plays a crucial role in the review. It requires significant revision.
Specifically, Table 1 lacks proper organization as the listed techniques are not on the same level or dimension for comparison. For example, 2-DE and Gel-free are presented at the same level, but Gel-free encompasses many of the mentioned methods. Furthermore, SILAC, TMT, and iTRAQ are labeling-based LC-MS methods and should be compared with label-free LC-MS methods. Another concern is that IAP is not generally considered a proteomics method, as it primarily quantifies only one or two proteins and leans more towards targeted analysis.
Moreover, there are several inaccuracies within the table that need to be addressed. For instance, MALDI does not exhibit high sensitivity and specificity for protein analysis; instead, it has considerably lower sensitivity for proteins compared to LC-MS. The author should consider reorganizing the table to enhance its logical flow and ensure accuracy by double-checking the information presented. Given that the entire review focuses on proteomics, this section requires significant improvement.
Furthermore, it is crucial to carefully consider which methods to include or exclude in the table. Inconsistencies arise later in the review where examples such as tissue microarray are not listed in the table, causing discrepancies. Please consider referring to reputable proteomics method reviews to assist in enhancing this section.
The authors could improve the readability of the review by making the sentences more concise and clear. For example, in lines 75-81, 116-121, and 125-132, there are lengthy sentences that can be shortened for better comprehension. Please also pay attention to typos and grammar throughout the review to ensure accuracy and professionalism.
Author Response
Dear Reviewer
Thank you for giving us the opportunity to submit a revised draft of our manuscript. We appreciate the time and effort that you and the other reviewers have dedicated to providing your valuable feedback on our manuscript. We are grateful for your insightful comments on our paper. We have been able to incorporate changes to reflect most of the suggestions provided by the reviewers. We have highlighted the changes within the manuscript.
Here is a point-by-point response to the reviewers’ comments and concerns.
R3
Comments and Suggestions for Authors
Overall, the review provides a comprehensive analysis and valuable insights into the future utilization of proteomics in the field of pancreatic cancer. However, there are concerns regarding the accuracy and organization of Table 1, which plays a crucial role in the review. It requires significant revision.
Specifically, Table 1 lacks proper organization as the listed techniques are not on the same level or dimension for comparison. For example, 2-DE and Gel-free are presented at the same level, but Gel-free encompasses many of the mentioned methods. Furthermore, SILAC, TMT, and iTRAQ are labeling-based LC-MS methods and should be compared with label-free LC-MS methods. Another concern is that IAP is not generally considered a proteomics method, as it primarily quantifies only one or two proteins and leans more towards targeted analysis.
Moreover, there are several inaccuracies within the table that need to be addressed. For instance, MALDI does not exhibit high sensitivity and specificity for protein analysis; instead, it has considerably lower sensitivity for proteins compared to LC-MS. The author should consider reorganizing the table to enhance its logical flow and ensure accuracy by double-checking the information presented. Given that the entire review focuses on proteomics, this section requires significant improvement.
Furthermore, it is crucial to carefully consider which methods to include or exclude in the table. Inconsistencies arise later in the review where examples such as tissue microarray are not listed in the table, causing discrepancies. Please consider referring to reputable proteomics method reviews to assist in enhancing this section.
Reply from authors: thank you for the comments, Table 1, which in this version of the manuscript became Table 2, was reorganized in order to make its information clearer.
We are also grateful to the reviewers for the identified imprecision that were corrected. Despite being extensive, the table in question intends to give readers with less experience in proteomics the potential of the different techniques available as well as the main problems that they may face if they intend to use them in their experiments.
Comments on the Quality of English Language
The authors could improve the readability of the review by making the sentences more concise and clear. For example, in lines 75-81, 116-121, and 125-132, there are lengthy sentences that can be shortened for better comprehension. Please also pay attention to typos and grammar throughout the review to ensure accuracy and professionalism.
Reply from authors: we thank the reviewer for the suggestions. All mentioned lines were corrected, and extended texts were transformed into more digestible and coherent sentences. Typos and grammar were identified and eliminated.

Round 2
Reviewer 2 Report
After initial comments, the authors made substantial changes to the manuscript. The current text is much better structured, more focused on proteomics in biomarker studies, and nicely presented with the addition of several tables. I recommend publication of the manuscript without modifications.
Reviewer 3 Report
The authors have addressed all the questions the reviewer had, and the reviewer does not have any additional comments. The reviewer is supportive of its publication in Proteomes.